# A Holistic Approach to IGBT Board Surface Fractal Object Detection Based on the Multi-Head Model

Haoran Huang and Xiaochuan Luo *

College of Information Science and Engineering, Northeastern University, No.11 Heping Region Wenhua Street, Shenyang 110819, China
* Correspondence: luoxch@mail.neu.edu.cn

**Abstract:** In industrial visual inspection, foreign matters are mostly fractal objects. Detailed detection of fractal objects is difficult but necessary because better decision-making relies on more detailed and more comprehensive detection information. This presents a challenge for industrial applications. To solve this problem, we proposed a holistic approach to fractal object detection based on a multi-head model. We proposed the IWS (Information Watch and Study) module to provide enhancement learning capabilities for object information. It increases the detection dimension of the object and can perform more detailed detection. In order to realize the portability of the IWS module, it can be easily and quickly deployed to the existing advanced object detection model to achieve end-to-end detection. We proposed the FGI (Fine-Grained Information) Head, which is used to extract more comprehensive feature vectors from the original base model. We proposed the WST (Watch and Study Tactic) Learner for object information processing and adaptive learning of class cluster centers. Using the MRD (Multi-task Result Determination) strategy to combine the classification results and IWS results, the final detection results are derived. In the experiment, the IWS and MRD were mounted on three different models of the YOLO series. The experimental results show that YOLO+IWS has good foreign object detection capabilities to meet the needs of industrial visual inspection. Moreover, for the detailed detection ability of fractal objects, YOLO+IWS is better than the other 11 competing methods. We designed a new evaluation index and an adjustment mechanism of class learning weights to make better judgments and more balanced learning. Not only that, we applied YOLO+IWS to form a brand new object detection system.

**Keywords:** fractal object detection; enhancement learning; fine-grained information; IGBT board





## 1. Introduction

Since the introduction of Industry 4.0, the traditional manufacturing industry is currently undergoing a transformation towards a digital, networked, and intelligent model. Intelligent manufacturing meets the demands of the personalized production market. This promotes the reorganization of production lines and the up-scaling of the process, which puts forward higher requirements for real-time performance, energy efficiency, and reliability of computing systems [1].

The detection of parts [2] and defects [3] is a crucial task in industrial visual inspection. In fields that require delicate work, it is indispensable to manually set the work area, shape, and posture for the target device. Furthermore, there is labor cost and inconvenience as the configuration has to be manually updated every time a new class of objects is detected. To automate these complex operations, real-time and efficient object detection methods are essential [4]. Object detection based on deep learning has made a lot of contributions to this requirement.

A complete automated fault detection usually consists of two main steps: key device detection and failure mode identification [5]. The purpose of key device detection is to locate and extract objects from images with complex backgrounds. After narrowing

the search, the next step is to identify failure types and locate their exact location within the key device [6]. The more detailed the data detected by the industrial vision inspection, the more comprehensive the information obtained, and the better decisions made. Imagine the following scenarios. If the detection of defects or foreign objects can only determine whether there are or not in industrial vision inspection, it does not make much sense. This kind of two-class object detection cannot locate the failure cause at all. To locate the specific cause of the fault, it is necessary to conduct a secondary investigation manually. If there are some very similar objects in industrial visual inspection, the similarities and differences between them are hard to distinguish. Some false detections are bound to occur, which will lead to the wrong location of the problem. Decision-makers may make wrong decisions based on these conditions. Both of these situations are to be avoided in industrial visual inspection.

However, there were few more detailed detection methods for objects. Such studies in the industrial setting were even rarer. This put forward new requirements for existing industrial visual inspection tasks. Regarding the more detailed detection of the fractal object, there are two ways according to the current technology. The first way is to expand the fine-grained recognition, and add the regression task of object positioning on this basis. Fine-grained object detection is an extension of fine-grained image recognition that is more meaningful for industrial applications [6] (refer to Figure 1). It is difficult for general detectors to accurately localize and classify fine-grained objects because the feature reuse in them amplifies the conflict between "classification" and "localization" in object detection [7]. The accuracy of object localization for fine-grained detection directly affects the fine-grained classification of objects. So far, there were very few studies on fine-grained object detection tasks that require both fine-grained object localization and classification. It can be seen that the research on this part is very difficult. The second way is to optimize the object detection method to strengthen the processing of fine-grained information. In this part of the research, how to extract and process fine-grained information is difficult.

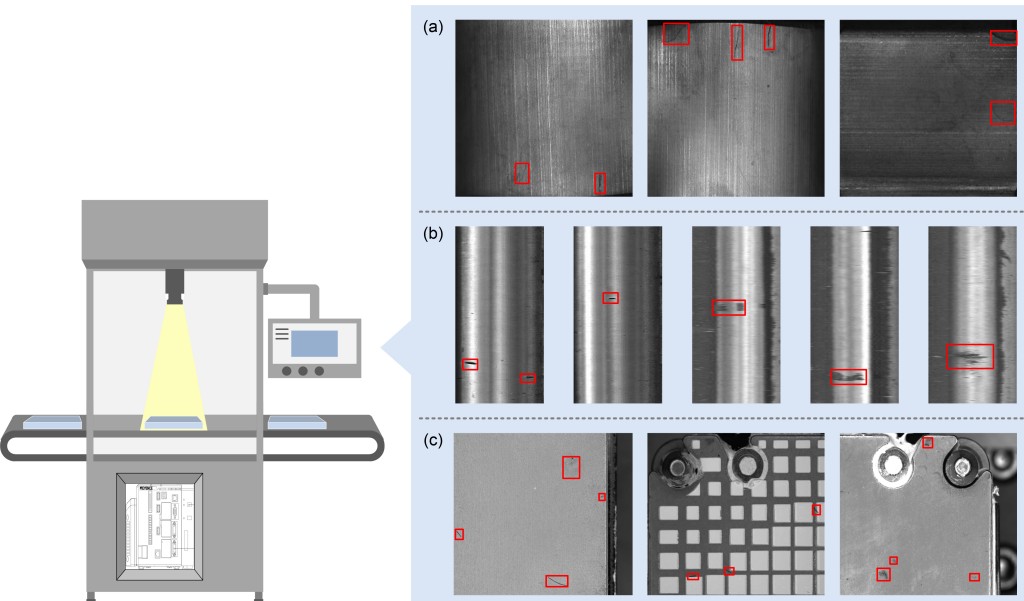

**Figure 1.** Objects in industrial vision inspection. (**a**) is the image data of the magnetic tile. (**b**) is the image data of the rail surface. (**c**) is the image data of the IGBT board surface. The characteristics of detection objects are very similar. It is necessary to carry out a more detailed detection of the object.

According to our experience, there are many uncertain factors in the first way. The first way is to face the following two problems. First, existing methods for fine-grained image recognition rely heavily on key local features. They are not suitable for scenes where the object cannot be split and the background feature has obvious rules (refer to Figure 2).

In the scenario where the object cannot be split, it is easy to scramble the reorganized graph to turn a single object into multiple small objects. It does not achieve the purpose of local feature extraction. On the contrary, it increases the difficulty of detection. In the scene with obvious regular background characteristics, the negative sample (background) area becomes more concentrated and the object becomes more sparse after being split into sub-areas. In particular, this phenomenon is more obvious when the object (such as hair) occupies a smaller area in the candidate box. Among classical methods of fine-grained image recognition, CAP [8], TransFG [9], PMG [10] and WS -DAN [11] are not suitable for the above two scenarios. DCL [12] and LIO [13] are more serious. Second, publicly natural datasets used to train models for fine-grained image recognition do not require separate object localization (refer to Figure 2). However, in practical industrial applications, it is difficult to extract sample data similar to public natural datasets. If we desire to make a dataset similar to publicly available natural datasets, there are two ways. The first is to use the preprocessing network model to regress the sample data of multiple objects into the sample data of multiple individual objects. This is very difficult. There are many uncertainties in this way. With this approach, it is difficult to achieve end-to-end detection, and it is not suitable for training or incremental learning. The second is to artificially collect a large number of sample data of individual objects, which is obviously not smart enough, and is not suitable for incremental learning. The method of the second way is more widely used in the industry than the first way, so the second way is relatively more meaningful.

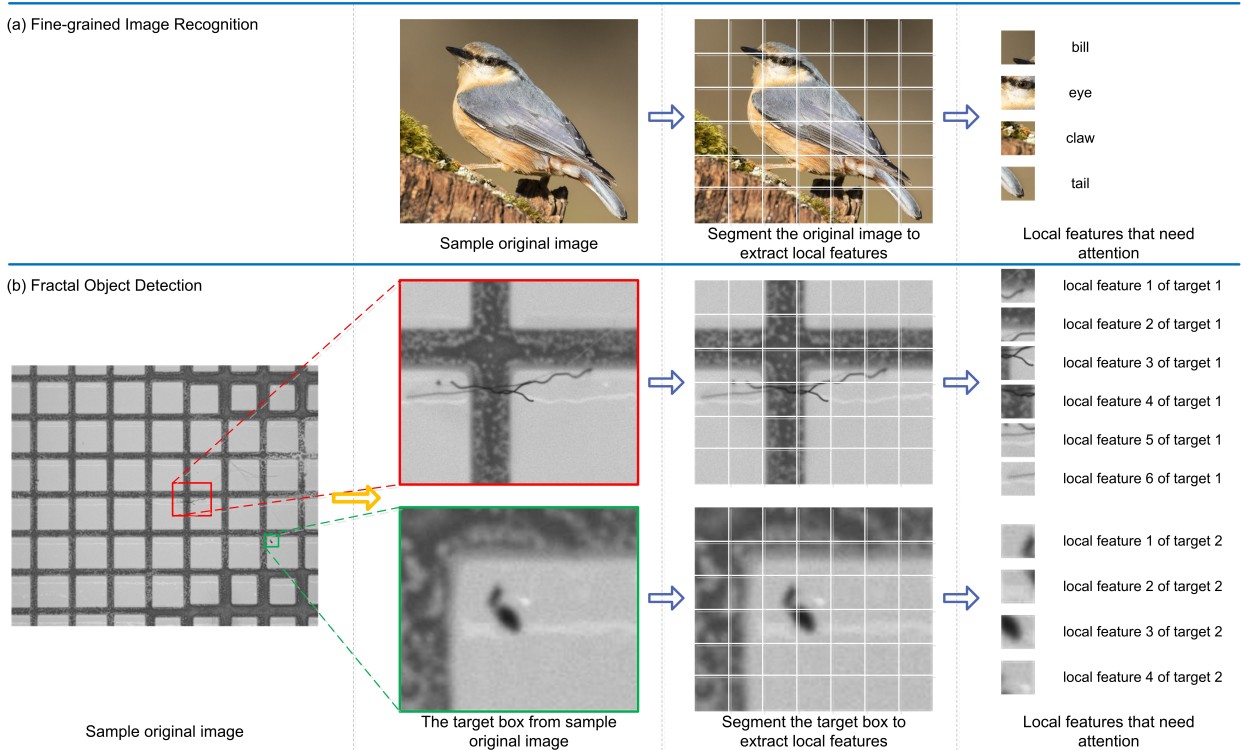

**Figure 2.** Fine-grained Image Recognition (**a**) and Fractal Object Detection (**b**).

In industrial visual inspection, foreign matters are mostly fractal objects. Fractal objects have a rough or fragmented geometric shape. It can be divided into several parts, and each part is a reduced shape of the whole. Because of these properties of fractal objects, it becomes difficult for multi-class detection [14]. It can be detected but is harder to subdivide. If you want to obtain detailed detection information, the detailed detection of fractal objects is indispensable in industrial visual inspection tasks.

Therefore, based on the existing advanced object detection models, we proposed a holistic approach to fractal object detection based on a multi-head model with object

information enhancement learning (refer to Figure 3). The main contributions of this paper are as follows:

1. The IWS (Information Watch and Study module) module (Section 3.1) was proposed by us to increase the detection dimension of object information;
2. We designed the FGI (Fine-Grained Information) Head (Section 3.1.1) to extract more comprehensive feature vectors. For object information calculation and class cluster center learning, we proposed a WST (Watch and Study Tactic) Learner (Section 3.1.2);
3. The MRD (Multi-task Result Determination) strategy (Section 3.1.3) that combines classification information and fine-grained information to give detection results were designed. We proposed an adjustment mechanism of class learning weights (Section 3.3). Its goal is to force the network model to fully learn the characteristics of each class. A new evaluation index (Section 3.2) was designed to facilitate better judgment.

This method can play a good role in the multi-head model. Under the edge-side image anomaly detection data, the new detection method formed by IWS on 3 different models of the YOLO series was compared with the advanced object detection models. The new detection method with IWS shows good results (refer to Figure 4).

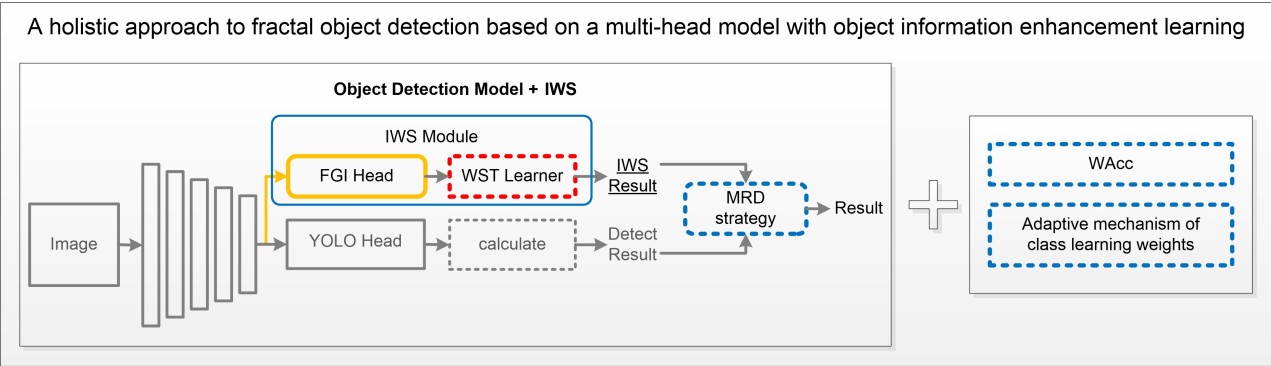

**Figure 3.** A holistic approach to fractal object detection based on a multi-head model with object information enhancement learning.

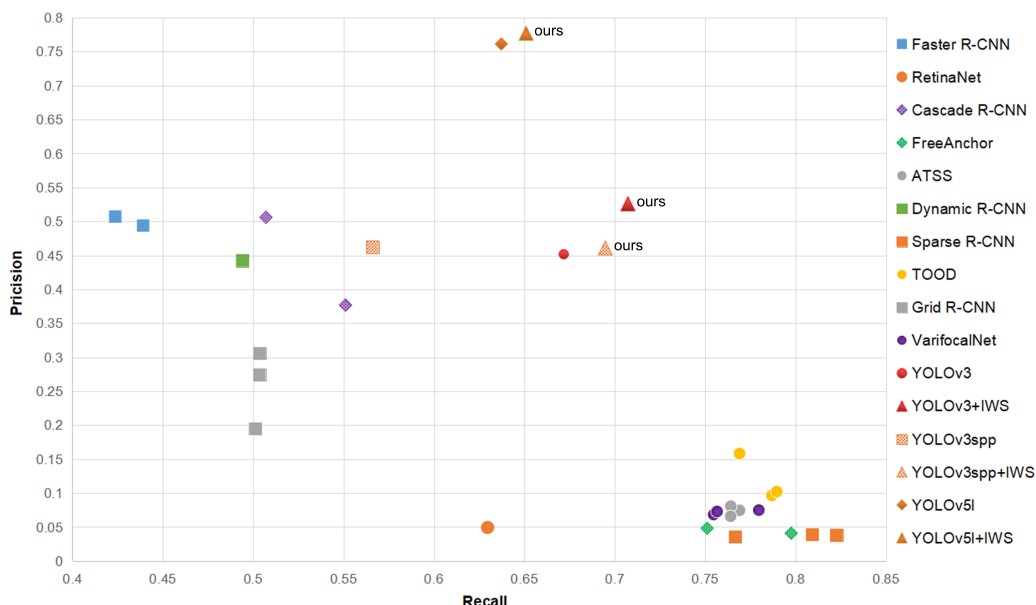

**Figure 4.** Our models have better effect than other competitive methods.

## 2. Related Work

Commonly known detection objects in the industry include fiber, hairs, packaging shavings, metal shavings, and scratches. It can be seen that most of these foreign objects have fractal characteristics. In the past, only a two-class detection method for detecting the presence or absence of foreign objects was required. However, this only meets basic needs; it can not provide instructive information. Industrial production lines in digital, networked, and intelligent modes need to know more and obtain more detailed inspection information. Then, it is necessary for the detailed detection of these fractal objects. However, fractal objects are difficult to be subdivided due to their nature, so the detailed detection of fractal objects is a challenging task.

### 2.1. Object Detection

In industrial visual inspection, traditional inspection methods were usually used. As time goes on, computing performance has been greatly improved. Deep learning methods are gradually replacing traditional object detection methods in the field of object detection. Deep learning methods have been widely used in various fields [15–20].

Different from traditional object detection algorithms, deep learning object detection algorithms are based on CNN (Convolutional Neural Network). CNN can automatically learn the features of objects through existing data. It can adapt to diverse backgrounds and object classes [21]. CNN-based object detection algorithms can be divided into two series from the perspective of network architecture. One was the object detection algorithm based on candidate regions represented by Fast-RCNN (Region-based Convolutional Neural Network) [22], Faster R-CNN [23], Mask R-CNN [24], etc. This type of algorithm usually had outstanding detection accuracy, but it had high requirements for the quality of the candidate box and the short slab of the detection speed. The other type was the regression-based object detection algorithm represented by YOLO (You Only Look Once) [25], SSD (Single Shot Multi-box Detector) [26], YOLO9000 [27], YOLOv3 [28], YOLOv4 [29], etc. The advantage of this kind of algorithm was a pleasurable real-time performance.

In addition, there were some excellent object detection models. Some object detection models were constructed around the Focal Loss function with the ability to mine hard examples. Representative networks were RetinaNet [30], Grid R-CNN [31], ATSS [32], Dynamic R-CNN [33], Sparse R-CNN [34] and VarifocalNet [35]. The other part of object detection models optimized around anchor free, e.g., Cascade R-CNN [36], FreeAnchor [37] and TOOD [38]. These excellent object detection models performed quite well under natural datasets. However, they might not be applicable in industrial application scenarios, especially for fractal object detection tasks.

Product inspection based on computer vision has been widely researched and applied. Yun et al. [39] proposed a conditional convolutional variational autoencoder (CCVAE) to generate images of different classes of metal defects, and designed a classifier using a deep convolutional neural network (DCNN) with high generalization. This method yielded excellent results for defect detection in actual metal production lines. However, the aforementioned cases could only be used for one class of object problems [40]. Chen et al. [41] proposed a YOLOv3-dense network by replacing Darknet-53 with DenseNet-121. It was used to detect the misplacement, missing wires, and surface defects of surface-mounted device light-emitting diodes (SMD LEDs). Zheng et al. [42] proposed an improved YOLOv3 network model that contains four submodels: the bottleneck attention network (BNA-Net), the attention prediction subnet model, the defect localization subnet model, and the large-size output feature branch. The network model could improve the recognition accuracy of large and medium defect objects on the bearing cover. Duan et al. [43] proposed a method that incorporates dual-density convolutional layers into YOLOv3 and expanded three feature maps of different scales in YOLOv3 to four. Yu et al. [44] proposed a separable residual module based on deep separable convolutions and residual networks. A network with shallower layers and fewer channels was designed for quick detection and recognition of commutator surface defects. It was applied to commutator surface defect detection

and recognition. Yao et al. [45] combined the proposed overlapping pooling spatial attention module and the dilated convolution module and applied it to high-precision and real-time inspection for the online defect detection of PAD (portable Android device) LGPs (Light guide plates). Tzab et al. [46] first used improved Yolov3-tiny to extract the object's cutting edge region. Then, the traditional image processing method was used to detect and evaluate defects. Obviously, YOLOv3 was widely used in industrial visual inspection applications. It was a favorable object detection model. However, it is still necessary to provide targeted optimization schemes for the characteristics of different detection objects. In particular, classes of objects were not included in natural datasets and the object has insignificant features.

### 2.2. Fractal Object Recognition

Refs. [47–50] conducted an applied study on the detailed detection of fine-grained fractal objects. However, these methods only solved the problem of recognition, not the problem of detection. While these methods do an extension on detection, no end-to-end optimization was given. In industrial scenarios, it was difficult for the two-stage object detection method to meet the industrial detection time requirements.

We needed to solve this problem and made an end-to-end enhancement learning detection method that can detect objects in more detail. Therefore, we proposed a holistic approach to fractal object detection with object information enhancement learning.

### 3. Methodology

The network structure of YOLOv3 mainly includes three parts: a Darknet-53 network, a functional Neck, and a YOLO Head layer. Darknet-53, as a backbone network, is mainly used to extract image features. Darknet-53 is a fully convolutional network that contains 53 convolutional layers and introduces a residual structure. When the input image size is 416 × 416, the Darknet-53 feature extraction network outputs feature maps of three scales. Their sizes are 13 × 13, 26 × 26, and 52 × 52. Three feature maps of different scales are processed by the functional Neck. Additionally, multiscale strategies are used to help the network model learn different levels of feature information at the same time. Finally, fused features are the input to the YOLO Head layer for class prediction and bounding box regression.

### 3.1. YOLO with IWS

In order to achieve end-to-end detection methods that can perform more detailed detection of fractal objects. We proposed an easy-to-deploy IWS module that provides enhancement learning capabilities on object information. It adds a detection dimension to the object and can perform more detailed detection. The explanation was based on YOLOv3 as the base model. Use it to carry our method, which we call YOLOv3+IWS below (refer to Figure 5).

We can deploy the target information enhancement learning module without changing the basic model structure and algorithm. We solved 3 difficulties. They were how to extract more comprehensive object information, how to process object information to achieve the purpose of enhancing the detection dimension, and how to balance the enhancement learning detection results of object information and the original basic network detection results. To this end, we proposed FGI Head to extract more comprehensive feature vectors and obtain more detailed object information. We proposed WST Learner to analyze the fine-grained information of the object and used prior knowledge to continuously accumulate learning. Combining the two parts of FGI Head and WST Learner, we proposed an object information enhancement learning module, which is called the IWS Module. In order to make the IWS Module easy to deploy, we proposed a multi-task result determination strategy, which is convenient for the model to integrate the multi-task results, so as to give better judgment results.

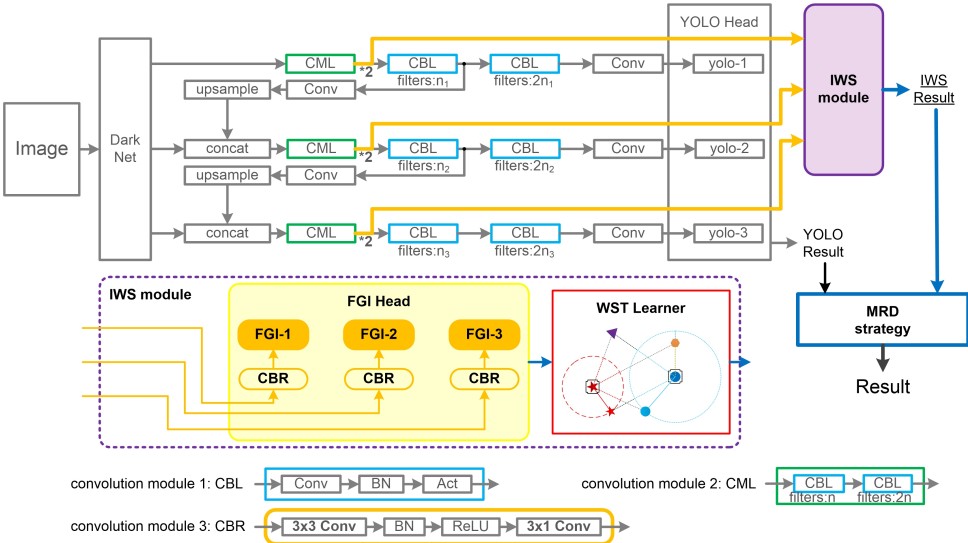

**Figure 5.** Core strategy of YOLO+IWS.

### 3.1.1. FGI Head

To perform a more detailed analysis of the fractal object, it was not enough to rely only on the feature vectors extracted by the classifier. The purpose of the classifier was to extract key features. The classifier used them to provide attribution probability about the class [28]. This treatment would only magnify the most critical feature and magnify the differences between classes. It might be difficult to do a more fine-grained analysis. Therefore, we needed to extract the invariant information of the fractal object, the more comprehensive the information, the better. For easy deployment, we also needed to extract this information without changing the structure of the original base model.

It was difficult to determine which node to extract from and how. Our analysis found that the branch where YOLOv3 does upsample (point P in Figure 6) is a functional node, and the CBL (blue box) in front of this node (P) is used to refine the upsampling information and object features of this branch. The CBL (blue box in Figure 6) after the node (P) is focused on refining and analyzing object features of this branch. Judging from these, we thought that the node behind the CML (green box in Figure 6) was the comprehensive extraction node of the object feature of this branch, which contains more comprehensive information. So we set up FGI Head at this node and added a $3 \times 1$ Conv (yellow box in Figure 6) which is used to extract the invariant information of object features for comparison. Therefore, we finally set up a new FGI Head without changing the original YOLO Head structure. It can well extract more comprehensive fractal object invariant information and was easy to insert.

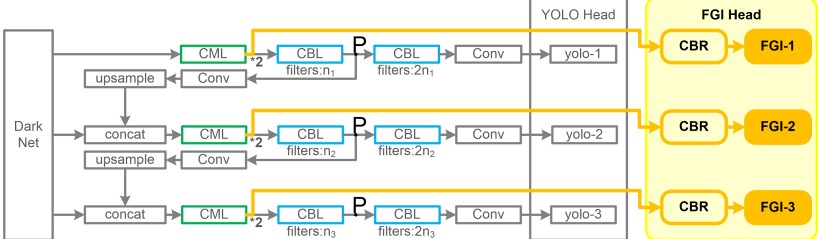

**Figure 6.** YOLOv3 with FGI Head module.

### 3.1.2. WST Learner

By deepening and widening the network, the high-dimensional feature vector was reduced in dimension to obtain a limited-dimensional feature vector. These feature vectors were compared item by item. The similarities and differences between the feature vectors were counted according to the difference of each feature item, and then the classification

result was obtained. This was the general idea for a more detailed detection of objects. We did not deal with it that way. Since the basic network has already done effective classification training, it was not meaningful to do a similar operation on the fractal object one more time. Although more comprehensive object feature invariant information was used as input, doing one more routine classification training by widening and deepening the network would only increase the difficulty of training.

It made sense to analyze the same object from different dimensions. Our approach was to map more comprehensive object feature invariant information into a fine-grained space. In this space, we did not need to do an item-by-item comparison of feature vectors, but to perform a watch and study tactic. WST Learner should not only make the feature vectors of the same class as close as possible, but also keep them as far away as possible from different classes. In addition, WST Learner can continue to learn along with the task and improve the detection ability.

Conventional classifiers consisted of multiple logical classifiers upon which to determine classification results. That is to say, the classifier could only distinguish the object class, and did not pay attention to the distribution of the feature vector in the feature space. If two objects were fine-grained fractal objects, their respective feature vectors were far away from other classes in the feature space, but most likely they were very close. This probability was higher especially when using Euclidean distance as a metric. We needed to avoid this in order to achieve a more detailed detection of the fractal object. Therefore, we made each YOLO Head branch correspond to its own FGI Head, and cluster the feature vectors of the same class label to obtain the feature vector of the class cluster center. Using this cluster center feature vector as the class calibration of the WST learner. The feature vector obtained by FGI Head was compared with all class cluster centers. The metric function is as follows:

$$d_{pre}^{(i)} = \begin{cases} I_{c(k)}^{(i)} \cdot \mathrm{CE}\left(F_{c(k)}, x^{(i)}\right), & \text{if } d_y^{(i)} = 0, i \in (0, N) \\ \left(1 - I_{c(j)}^{(i)}\right) \cdot \mathrm{CE}\left(F_{c(j)}, x^{(i)}\right) - I_{c(k)}^{(i)} \cdot \mathrm{CE}\left(F_{c(k)}, x^{(i)}\right), & \text{if } d_y^{(i)} = 1, i \in (0, N), k \neq j. \end{cases} \tag{1}$$

$I_{c(k)}^{(i)}$ represents the decision regarding whether or not $x^{(i)}$ belongs to class $k$. When $d_y^{(i)}$ is 0, the calculation is the metric of the feature vector $x^{(i)}$ and the cluster center of the belonging class. When $d_y^{(i)}$ is 1, the difference between the measure of the feature vector $x^{(i)}$ in the cluster and the measure of the distance from the cluster center of other classes is calculated separately.

Cosine distance was concerned with the overall distribution over all dimensions. It can better reflect the spatial distribution of high-dimensional features. It could very well reflect the degree of similarities and differences between the directions of the vectors [51] ($Sim(a,b) = \frac{a \cdot b}{||a|| \cdot ||b||}$). It could provide a new dimension to analyze object information. However, cosine distance was somewhat insufficient. A vector in the same direction could not reflect the distance relationship (as shown in Figure A2). We needed a method that can focus on both the overall spatial distribution and the distance differences between vectors under the same distribution. So we optimized it. It was used in metric calculation. The formula is as follows:

$$CE(a,b) = (a - b)_{L_2} \cdot (1 - Sim(a,b)). \tag{2}$$

It was not difficult to imagine that the determination of class cluster centers would directly affect the classification effect. If the class cluster center was fixed, it was required to be accurate enough, which is obviously difficult. Then, we desired to make the dynamic class cluster center. It could be less accurate at first but evolved as it learned. It is a class cluster that keeps gaining experience. Inspired by the momentum gradient optimization method, we designed an evolutionary learning algorithm of the class cluster center.

During the training process, the class cluster center of each class would be continuously modified along with batches. The correction function is as follows. The class cluster center counted by the current batch refers to the prior knowledge of the class cluster center that has been accumulated and learned before obtaining the class cluster center used for contrastive learning in the current batch. $\alpha$ represents the degree of contribution. In the reasoning process, $\beta$ is greater than $\alpha$ in terms of contribution. $F_{c(k)}^{b(l)}$ represents the feature vector of the cluster center belonging to class $k$ under batch $l$. $\alpha_{c(k)}$ represents the contribution degree of cluster centers inheritance belonging to class $k$ in the training phase. $\beta_{c(k)}$ represents the inheritance contribution degree of cluster centers belonging to class $k$ in the inference stage. $f_{c(k)}$ indicates that the feature vector of the cluster center belonging to class $k$ is counted under the current batch. Watching and analyzing the relationship between the feature vector of each batch of objects and cluster centers of classes provided a basis for the class judgement of the object. Based on prior knowledge, learning and optimizing the expression for cluster centers of classes. This idea is our watch and study tactic. Based on this, the WST Learner was formed (refer to Figure 7) (see Algorithm 1). The formulas are as follows:

$$F_{c(k)}^{b(l)} = \alpha_{c(k)} \cdot F_{c(k)}^{b(l-1)} + \left(1 - \alpha_{c(k)}\right) \cdot f_{c(k)}, \text{ if training} \tag{3}$$

$$F_{c(k)} = \beta_{c(k)} \cdot F_{c(k)}^{b(L)} + \left(1 - \beta_{c(k)}\right) \cdot f_{c(k)}, \text{ if inference.} \tag{4}$$

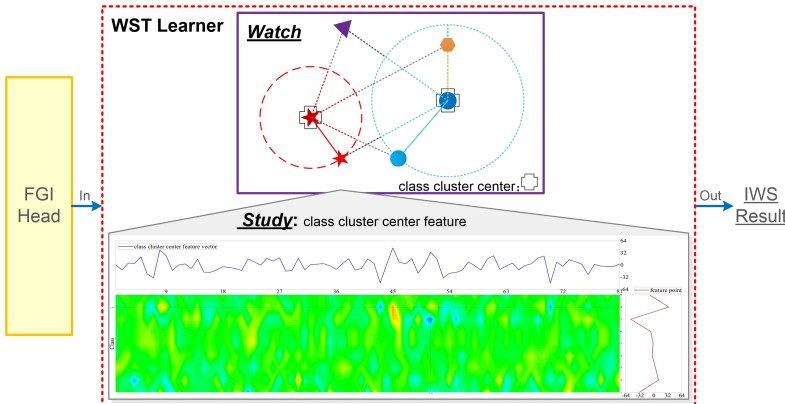

**Figure 7.** WST Learner calculation module.

---

**Algorithm 1** WST Learner in training phase.

---

**Input:** batch number $l \in (0, L)$, class number $k \in (0, K)$, the number of feature vectors
　　　belonging to class k is $N_{c(k)}$, class cluster centre feature vector $F = \left\{F_{c(0)} \cdots F_{c(K)}\right\}$,
　　　feature number $i \in (0, N_{c(k)} - 1)$, feature vector $x$
**Output:** WST Loss $Loss_{WST}$
　1: **if** $l == 0$ **then**
　2:　　Initialize $F_{c(k)}^{b(l)} \leftarrow \{0...0\}$
　3: **else**
　4:　　**if** $N_{c(k)}^{b(l)} == 1$ **then**
　5:　　　　$F_{c(k)}^{b(l+1)} \leftarrow F_{c(k)}^{b(l)}$
　6:　　**else**
　7:　　　　$F_{c(k)}^{b(l+1)} \leftarrow Compute \text{ using } F_{c(k)}^{b(l)}, f_{c(k)}^{b(l)}$ and Equation (3)
　8:　　**end if**
　9: **end if**
10: $d_{pre}^{(i)} \leftarrow Compute \text{ using } F, x^{(i)}$ and Equation (1)
11: $Loss_{WST} \leftarrow Compute \text{ using } d_{pre}^{(i)}$ and Equation (5)
12: **return** $L_{WST}$

---

Based on the metric algorithm and the evolutionary learning algorithm for the class cluster center, the loss function was finally designed as follows:

$$Loss_{WST} = -(1/N) \sum_{i=1}^{N} \left( d_y^{(i)} \log\left(D_{pre}^{(i)}\right) + \left(1 - d_y^{(i)}\right) \log\left(1 - D_{pre}^{(i)}\right) \right). \tag{5}$$

$$D_{pre}^{(i)} = \begin{cases} At\left(\lambda_1 \cdot Bn(d_{pre}^{(i)})\right), & \text{if } d_y^{(i)} = 1 \\ At\left(\lambda_0 \cdot (Bn(d_{pre}^{(i)}) - 1)\right), & \text{if } d_y^{(i)} = 0. \end{cases} \tag{6}$$

When $d_y^{(i)}$ is 0, $d_{pre}^{(i)}$ is a value greater than 0. The smaller the value is, the better. When $d_y^{(i)}$ is 1, $d_{pre}^{(i)}$ is any value. The larger the value, the better. When a negative value appears, it means that the current sample is mixed into a class that it does not belong in, and a serious penalty is required. $At(.)$ is the activation function. We were using Sigmoid here.

### 3.1.3. Multi-Task Result Discriminant Strategy

The classifier that came with the YOLO series model produced the classification judgment result. The IWS module we proposed also produced a judgment result. The model needed to weigh two outcomes. The results of two analyses of different dimensions could be viewed as two independent tasks. Accordingly, we designed a multi-task result discriminant strategy to help the model give an accurate final conclusion (see Algorithm 2). This also helped the model accumulate data samples for incremental learning. It provided a basis for the model to discover new classes of objects (or unknown classes).

If the result of the classifier is inconsistent with the IWS result and the object's low score from the classifier, it can basically be determined as an unknown class object. $R_{c(k)}$ is the cluster radius of class $k$. It is obtained by accumulating learning through model training. $O_{c(k)}^{(i)}$ represents the metric from the current object $i$ to the cluster center of class $k$. $O^{(i)}$ is the set of metrics from the current object $i$ to the cluster centers of each class. Suppose $c(i)$ and $c(j)$ are the smallest metric difference and the next smallest metric difference in $O^{(i)}$, respectively.

$$O_{c(k)}^{(i)} = CE\left(F_{c(k)}, x^{(i)}\right) - R_{c(k)}. \tag{7}$$

### 3.2. WST Accuracy (WAcc)

To better observe and evaluate our optimization utility, we set an evaluation index. WAcc (WST Accuracy) represents the probability of correct distribution in the feature vector space and the correct final classification, reflecting the ability to distinguish similarities and differences between the feature vector space and various class cluster centers. $M$ represents an extremely large number. The purpose of this is to make its value result tend to 0 or 1. $\xi$ is the threshold for judging similarities and differences. For all classes, WAcc counts the ability to distinguish similarities and differences of all classes.

$$WAcc = (1/N) \sum_{i=1}^{N} \frac{d_y^{(i)} + \left(1 - d_y^{(i)}\right) exp^{-M\left(d_{pre}^{(i)} - \xi\right)}}{1 + exp^{-M\left(d_{pre}^{(i)} - \xi\right)}}. \tag{8}$$

The WAcc indicator was different from the Acc (Accuracy) indicator. Acc reflected the accuracy of network model detection based on two classifications. The two-class judgement that decides yes or no could be used to determine the effect of contrastive learning. However, Acc was somewhat general. Contrastive learning results in the case of multiple classifications could not be reflected under Acc. Therefore, we optimized Acc. Combined with the design idea of IWS module, the IWS learning effect was combined with the classification effect to form WAcc. Such a design could not only reflect the

effect of contrastive learning but also reflected the effect of accurate classification under contrastive learning.

---

**Algorithm 2** Multi-Task Result Discriminant strategy.

---

**Input:** candidate results of the IWS module for object $i$ are $J_{c(i)}^{(i)}$ *and* $J_{c(j)}^{(i)}$, $O_{c(i)}^{(i)}$ *and* $O_{c(j)}^{(i)} \in$

　　　　$O^{(i)}$, result of the YOLO classier is $Y_{c(k)}^{(i)}$

**Output:** result of the model classier is $Cls^{(i)}$

　1: **if** $O_{c(i)}^{(i)} \neq O_{c(j)}^{(i)}$ **then**

　2:　　**if** $O_{c(i)}^{(i)} \geq 0$ **then**

　3:　　　　**if** $Y_{c(k)}^{(i)} \neq J_{c(i)}^{(i)}$ **then**

　4:　　　　　　$Cls^{(i)} \leftarrow$ new or unkonw class. Classify $i$ as novelty samples

　5:　　　　**else**

　6:　　　　　　$Cls^{(i)} \leftarrow Y_{c(k)}^{(i)}$

　7:　　　　**end if**

　8:　　**else**

　9:　　　　**if** $Y_{c(k)}^{(i)} \neq J_{c(i)}^{(i)}$ **then**

　10:　　　　　Classify $i$ as hard samples

　11:　　　　**end if**

　12:　　　$Cls^{(i)} \leftarrow Y_{c(k)}^{(i)}$

　13:　　**end if**

　14: **else**

　15:　　**if** $O_{c(i)}^{(i)} \geq 0$ **then**

　16:　　　　**if** $Y_{c(k)}^{(i)} \neq J_{c(i)}^{(i)}$ *and* $Y_{c(k)}^{(i)} \neq J_{c(j)}^{(i)}$ **then**

　17:　　　　　　$Cls^{(i)} \leftarrow$ new or unkonw class. Classify $i$ as novelty samples

　18:　　　　**else**

　19:　　　　　　$Cls^{(i)} \leftarrow Y_{c(k)}^{(i)}$. Classify $i$ as hard samples

　20:　　　　**end if**

　21:　　**else**

　22:　　　　**if** $Y_{c(k)}^{(i)} \neq J_{c(i)}^{(i)}$ *and* $Y_{c(k)}^{(i)} \neq J_{c(j)}^{(i)}$ **then**. Classify $i$ as hard samples

　23:　　　　**end if**

　24:　　　$Cls^{(i)} \leftarrow Y_{c(k)}^{(i)}$

　25:　　**end if**

　26: **end if**

　27: **return** $Cls^{(i)}$

---

### 3.3. Adjustment Mechanism of Class Learning Weights

Through observation, we found that the network model was more inclined to learn classes that are easy to learn because it was easier to obtain high-quality evaluation indicators. However, those difficult classes will be ignored and even tend to overfit. This phenomenon occurred in the vast majority of network models. In the practical application of industrial visual inspection, the collected sample data were very likely to have an unbalanced number of classes. This situation increased the probability of the occurrence of the above phenomenon. Therefore, some optimizations in this area were needed to avoid this phenomenon. To allow the network model to fully learn the characteristics of each class, we designed the adjustment mechanism of class learning weights. The situation where a network model learns only easy-to-learn classes and abandons difficult-to-learn classes is avoided.

For multiple classes of objects, different classes were given their own learning weights. After each full batch of data (epoch) training, a statistical analysis of the effect of each class of training was performed. The network model refers to the learning effect of this time and gives learning weights for each class of objects to the next epoch training (see Algorithm 3).

In this way, targeted adaptive learning can be achieved, thereby improving the effect of network model training.

---

**Algorithm 3** Adjustment mechanism.

---

**Input:** class weights $W = \left\{ \omega_{c(0)} \cdots \omega_{c(K)} \right\}$, class Loss $Loss = \left\{ loss_{c(0)} \cdots loss_{c(K)} \right\}$, epoch size $T$, epoch number $t \in (0, T)$, momentum parameter of class learning is $\eta$

**Output:** $W$
1: **if** t == 0 **then**
2:     Initialize $\omega_{c(k)}^{t} \leftarrow 1$ and $\eta \leftarrow 0.9$
3: **end if**
4: **if** t > 0 **then**
5:     $m_{c(k)}^{t} \leftarrow loss_{c(k)}^{t} / \sum_{i \in (0, K)} loss_{c(i)}^{t}$
6:     $\omega_{c(k)}^{t+1} \leftarrow \eta \omega_{c(k)}^{t} + (1 - \eta) m_{c(k)}^{t}$
7:     $\text{Loss}_{cls} \leftarrow \sum_{i \in (0, K)} \omega_{c(i)}^{t+1} loss_{c(i)}^{t+1}$
8: **end if**
9: **return** $W$

---

In the test phase of epoch $t$, each class object calculates its own loss function. Based on these losses, we can obtain the momentum offset $m_{c(k)}^{t}$ of each class object. The larger the learning deviation at epoch $t$ is, the larger $m_{c(k)}^{t}$ is. Then, the class weight $\omega_{c(k)}^{t+1}$ (shown in step 6 in Algorithm 3) assigned to the training stage of epoch $t + 1$ will be adjusted more. This means that the network model needs more learning about the class $c(k)$ in epoch $t + 1$ than epoch $t$. We use $\eta$ to control the range of weight change and avoid the sudden change of weight in the whole process.

## 4. Approach

### 4.1. Experimental Dataset

To verify the effectiveness of YOLO+IWS, we chose edge-side image anomaly detection equipment as the application project. We conducted purposeful data collection in the IGBT automatic gluing operation line of Beijing Zongheng Electromechanical Co., Ltd.

Foreign object detection of the IGBT board needs to be performed twice under working conditions of a clean board and a coated board. We defined the collected dataset as the IGBT board surface object detection dataset (referred to as IGBT_DF). The sample we collected will be planned into seven classes of objects (corresponding objects shown in Figure 8), considering three perspectives:

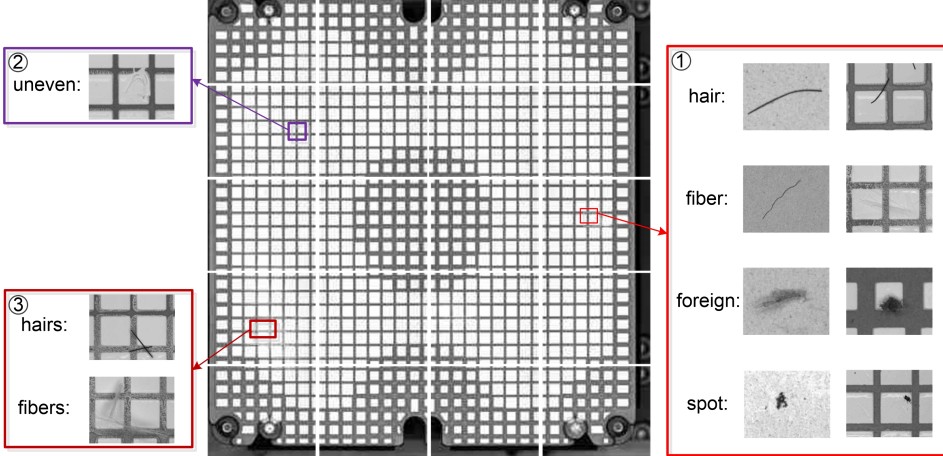

**Figure 8.** Seven classes of fractal objects in the IGBT Dataset.

1.  Common foreign object collection, including hair, fiber, packaging crumbs (foreign), and object crumbs (spot) are used for learning characteristics of common fractal objects. The main learning sample of the fractal object detectability for the dust-free future provides information support for the management and control of bacteria in the workspace (shown as ① in Figure 8). The purpose of doing this is not only to detect whether it is a foreign object, but also what kind of foreign object it is. So we need a multi-class classification dataset, not a binary classification dataset.

2.  Glue application collection, including uneven glue application (shown as ② in Figure 8). Since foreign and spot are similar, fiber and uneven are also difficult to distinguish (due to low contrast). We used foreign and spot, fiber and uneven as the main fractal object detection groups;

3.  The collection of complex objects, including cross hairs and fibers, is used to learn the characteristics of objects and enhance the model detectability (shown as ③ in Figure 8). Whether the detection network can effectively detect when the number of fractal objects changes in complex situations is investigated.

To ensure that the dataset built for the IGBT board surface object detection problem is useful and effective, our dataset borrows several designs from the natural dataset, MS-COCO [52]. Compared with the object distribution and proportion of the natural dataset, MS-COCO, some uncertain factors in our dataset that affect the training of the network model are eliminated. Such a processing method enables the network model to learn the characteristics of foreign objects more effectively and perform foreign object detection tasks better. Proportions of large-sized, medium-sized, and small-sized objects in our dataset IGBT_DF are similar to those in the natural dataset, MS-COCO, as shown in Table 1.

**Table 1.** Object distribution of the IGBT dataset.

| Data Set | At | Lt | Mt | St |
|---|---|---|---|---|
| MS-COCO | 3.5–7.7 | 25% | 34% | 41% |
| IGBT_DF (ours) | 4–8 | 27% | 27% | 46% |

At: Average number of objects in each sample. Lt: Proportion of the large size object (resolution $> 96 \times 96$) in the dataset. Mt: Proportion of the medium size object ($32 \times 32 <$ resolution $\leq 96 \times 96$) in the dataset. St: Proportion of the small size object ($32 \times 32 \geq$ resolution) in the dataset.

There is a definition here to clarify. There are two ways to define small objects. One is by the relative size. According to the definition of the small-sized object by the international organization SPIE, it is the object in the image with a relative size of 0.12% in the image [53]. The other way is by absolute size. According to the definition of the MS-COCO dataset, an object with a size smaller than $32 \times 32$ pixels can be regarded as a small object [52,54]. Based on the definition of relative size, the captured image resolution is $2432 \times 2040$, and then the small-sized object should not be larger than 5953 pixels. In terms of foreign object characteristics in the IGBT board surface object detection problem, this obviously does not meet the problem scenario of this article. Therefore, our dataset IGBT_DF adopts absolute size as the defining standard for small objects.

All images are randomly shuffled, 80% of which are divided into the train subset, and the rest are classified as test subset (as shown in Table 2). Figure 9 shows the fractal object distribution of the IGBT_DF dataset. The position of the fractal object in the image is evenly distributed. We used the classic *LabelImg* software to label the IGBT_DF dataset.

*4.2. Evaluation Metrics*

The evaluation criteria used in the COCO dataset are P (Precision), R (Recall), and AP (Average Precision). The larger the value of P is, the smaller the false detection rate. The larger the value of R is, the smaller the missed detection rate. AP is the area under the PR curve. P, R, and AP are for a single class. mAP (Mean Average Precision) is for all classes. The larger the value of mAP is, the better the overall performance of the learner for all classes of objects. Similarly, we used mP and mR to represent the mean of P and R of

multiple classes. In practical applications, Acc (Accuracy) is used as a reference indicator for the basic needs of the detection task which is used to evaluate the quality of samples in IGBT automatic glue coating operation. For IGBT board surface object detection, first of all, it is necessary to ensure that the basic requirement index (Acc) of object detection meets the standard to ensure the normal operation of the IGBT board surface object detection task. Then, the detailed detection capability of the fractal object in the IGBT board surface object detection task is investigated. The mAP index, specific indicators of missed detection, and false detection should be considered. Missed detection will result in missed warnings for foreign objects in the data analysis and statistics stage. False detection will result in false alarms for foreign objects. When mAP shows well, it is also indispensable to observe the conditions of the indicators mR and mP. As long as there is a foreign object, the sample must be recovered and reprocessed. WAcc (Section 3.2) reflects the ability of the detection model to discriminate between similarities and differences of objects. The larger the value of WAcc is, the stronger the discrimination ability, as follows (the meanings of $Tp$, $Fp$, $Tn$, and $Fn$ can be found in Table 3):

$$P = Tp/(Tp + Fn) \tag{9}$$

$$R = Tp/(Tp + Fp) \tag{10}$$

$$AP = \int P(R)dR \tag{11}$$

$$Acc = (Tp + Tn)/(Tp + Fn + Tn + Fp). \tag{12}$$

**Table 2.** The specific information of IGBT_DF dataset.

| IGBT_DF | Image | Object | Specific Information (Objects) | | | | | | |
|---|---|---|---|---|---|---|---|---|---|
| | | | **Hair** | **Hairs** | **Fiber** | **Fibers** | **Spot** | **Foreign** | **Uneven** |
| Train subset | 717 | 3018 | 739 | 58 | 395 | 84 | 722 | 883 | 137 |
| Test subset | 180 | 840 | 253 | 14 | 102 | 22 | 152 | 256 | 41 |

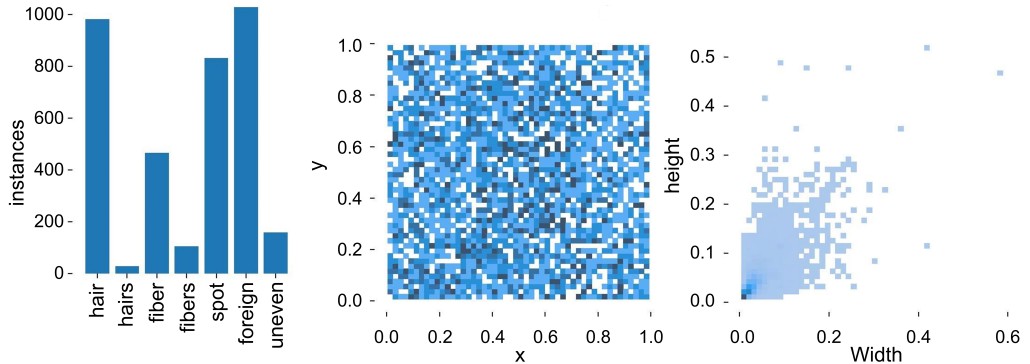

**Figure 9.** Fractal object distribution of the IGBT_DF: the number of various fractal objects (**left 1**), the position of the fractal object in the picture (**right 2**), and the size of the fractal object (**right 1**).

**Table 3.** Basic indicators for evaluation.

| | Positive | Negative |
|---|---|---|
| True | *Tp* | *Tn* |
| False | *Fp* | *Fn* |

### 4.3. Experimental Apparatus

Our network model finally needs to be mounted on the device server, configured with a CPU (Intel@ Xeon(R) CPU E5-262- v4 @2.10 GHz ×4), 16 GB RAM (random access memory, RAM) and GPU (NVIDIA Tesla P100 with 8 GB). Each IGBT board produces 20–25 sample pictures, and there is a small cross between samples. The resolution of each sample is 2432 × 2040. According to requirements of working conditions, the detection capability should be at least 30 fps and the object detection accuracy rate is not less than 88%.

### 4.4. Implementation Details

Depending on the edge-side device configuration, we used the Adam optimizer with a momentum of 0.9 and a learning rate of 0.001. GPU memory is 8 GB. The training strategy with the epoch of 300 was used to verify the optimization effect. This setting is more suitable for the actual use of field edge devices.

## 5. Results and Discussion

### 5.1. Experimental Analysis with Industrial Visual Inspection Data

#### 5.1.1. Compared with State-of-the-Art Approaches

The problem we desire to solve is the fractal object detection problem. The object detection model required more detailed detection of the object. The object that needs to be detected is the fine-grained object. Consequently, we decided to compare YOLOv3+IWS with excellent network models in the field of object detection. Looking at Table 4, under the same computing resources, through the IGBT_DF dataset, compared with an army of network models, YOLOv3+IWS has the best mAP and Precision. It is not difficult to see from FPS (frames per second) that YOLOv3 is more suitable for practical applications than other network models. Under the premise of ensuring that the detection capability is not lower than the 30fps working condition, YOLOv3+IWS is superior to YOLOv3 in all evaluation indicators. YOLOv3+IWS had a 2.14% improvement in mAP, and a 6.39% improvement in the similarities and differences identification indicator (WAcc). There is a 1.62% improvement in the reference indicator (Acc). In Recall, YOLOv3+IWS does not perform as well as Sparse R-CNN. However, other metrics of Sparse R-CNN are not so perfect. YOLOv3+IWS is more comprehensive. The basic object detection ability (Acc) of Sparse R-CNN is not as good as that of YOLOv3+IWS. Especially Precision, Sparse R-CNN has serious false detections. This will bring a lot of trouble to the IGBT gluing system. The experimental results show that the IWS module has indeed improved the detection ability of fractal objects more comprehensively by adding one more analysis dimension to object information by enhancement learning.

Under the IGBT_DF dataset, multiple network models exhibit high Recall but particularly low Precision. We considered this has nothing to do with the network structure. Through analysis and thinking, we found that network models with this phenomenon all use Focal Loss. Focal Loss is used in the image field to solve the model performance problem caused by data imbalance. Obviously, Focal Loss is not applicable to our dataset for objects. This is an interesting finding, but it is not what this study intends to discuss. It will not be discussed or extended here.

In addition, we separately listed models that meet the accuracy requirements of the IGBT board surface object detection task and have mAP exceeding 40%. These models were compared against AP for each class (refer to Figure 10). It can be found that YOLOv3+IWS improved on three pairs of objects, and is the best indicator on "hair", "hairs", "fiber", "spot" and "foreign". YOLOv3+IWS does not focus more on easy-to-learn objects. Avoid the phenomenon of brushing high mAP by using the object of learning easy to learn. Compared with other models, each class indicator under YOLOv3+IWS will be more uniform. YOLOv3+IWS can ensure that the overall performance improved, and each class can be relatively fully learned.

**Table 4.** Experimental results of IGBT_DF dataset.

| Model | Backbone | Acc | mAP | mR | mP | FPS |
|---|---|---|---|---|---|---|
| Faster R-CNN | Resnet-50 | 90.69% | 39.11% | 43.92% | 49.39% | 21.4 |
| | Resnet-101 | 90.74% | 35.86% | 42.38% | 50.71% | 15.6 |
| RetinaNet | Resnet-101 | 89.23% | 24.28% | 62.97% | 4.93% | 15 |
| Cascade R-CNN | Resnet-50 | 89.66% | 39.17% | 50.71% | 50.59% | 16.1 |
| | Resnet-101 | 90.09% | 39.44% | 55.11% | 37.67% | 13.5 |
| Grid R-CNN | Resnet-50 | 90.75% | 38.47% | 50.11% | 19.49% | 15 |
| | Resnet-101 | 91.27% | 37.15% | 50.35% | 27.44% | 12.6 |
| | Resnext-101 | 91.33% | 38.42% | 50.36% | 30.59% | 10.8 |
| FreeAnchor | Resnet-50 | 91.34% | 34.77% | 75.11% | 4.81% | 18.4 |
| ATSS | Resnet-50 | 90.92% | 48.50% | 76.90% | 7.46% | 19.7 |
| | Resnet-101 | 91.26% | 43.14% | 76.42% | 8.07% | 12.3 |
| | Resnext-101 | 91.34% | 43.7% | 76.41% | 6.58% | 11 |
| Dynamic R-CNN | Resnet-50 | 89.59% | 40.84% | 49.40% | 44.24% | 18.2 |
| Sparse R-CNN | Resnet-50 | 89.33% | 35.12% | **82.26%** | 3.83% | 22.5 |
| | Resnet-101 | 89.86% | 38.93% | 76.66% | 3.57% | 18.5 |
| | Resnext-101 | 89.92% | 39.37% | 80.92% | 3.91% | 17 |
| TOOD | Resnet-50 | 90.04% | 47.05% | 76.90% | 15.83% | 19.3 |
| | Resnet-101 | 90.53% | 44.40% | 78.69% | 9.66% | 18.1 |
| | Resnext-101 | 90.12% | 47.67% | 78.96% | 10.21% | 17 |
| VarifocalNet | Resnet-50 | 90.67% | 47.01% | 77.97% | 7.49% | 19.3 |
| | Resnet-101 | 90.56% | 47.24% | 75.47% | 6.84% | 15.6 |
| | Resnext-101 | 90.51% | 47.67% | 75.66% | 7.29% | 14 |
| YOLOv3 | Darknet-53 | 89.86% | 50.98% | 67.18% | 45.16% | **35** |
| YOLOv3+IWS (ours) | Darknet-53 | **91.48%** | **53.12%** | 70.73% | **52.58%** | 34 |

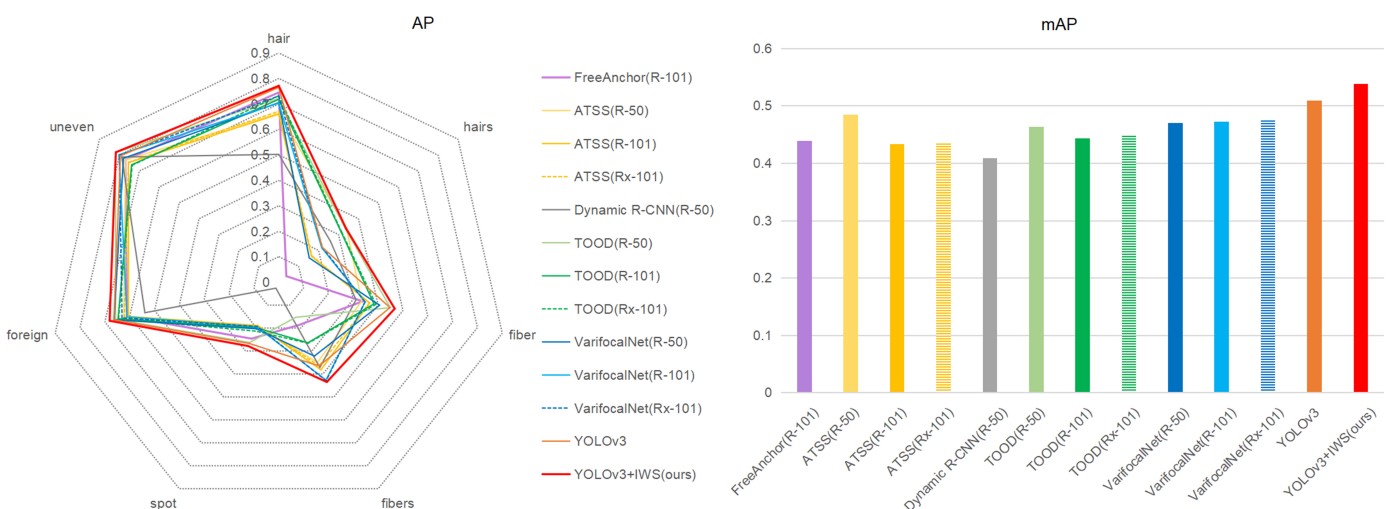

**Figure 10.** Comparison of AP values for each class of models with mAP exceeding 40% (R-50: Resnet-50, R-101: Resnet-101, Rx-101: Resnext-101).

With the increase in operation time, the data continue to accumulate. Edge-side devices need to use the increasing data to improve their detection capabilities. It is worth considering whether the detection model could steadily improve the ability. We designed a larger dataset, which we defined as IGBT_DF_L (shown in Table 5 and the specific information be found in Figure A3). Under the IGBT_DF_L dataset, we examined the learning situation of the network model under large quantities of data and observed the improvement effect of mAP. First, it can be seen that all network models will have a certain

degree of improvement when the amount of data increases. This demonstrates that the dataset we built is effective, and can be used for training and learning characteristics of objects. YOLOv3+IWS consistently shows an excellent result, mAP has been improved (shown in Table 6).

**Table 5.** Object distribution of IGBT_DF_L dataset (More experimental results can be found in Table A1) for training.

| IGBT_DF_L | Train Images | Test Images | Train Objects | Test Objects |
|---|---|---|---|---|
| | 3544 | 886 | 18,139 | 4512 |

**Table 6.** Experimental results of IGBT_DF_L dataset.

| Model | FRC | | RN | CRC | | GRC | | | FA | |
|---|---|---|---|---|---|---|---|---|---|---|
| | R-50 | R-101 | R-101 | R-50 | R-101 | R-50 | R-101 | Rx-101 | R-50 | R-101 |
| mAP | 44.09% | 43.71% | 43.25% | 45.09% | 44.98% | 42.82% | 42.70% | 43.17% | 52.50% | 52.31% |

| Model | ATSS | | | DRC | | SRC | | | TOOD | |
|---|---|---|---|---|---|---|---|---|---|---|
| | R-50 | R-101 | Rx-101 | R-50 | R-50 | R-101 | Rx-101 | R-50 | R-101 | Rx-101 |
| mAP | 52.80% | 52.10% | 52.34% | 45.12% | 44.84% | 53.33% | 54.11% | 54.01% | 52.15% | 53.17% |

| Model | VN | | | Y3 | Y3-T |
|---|---|---|---|---|---|
| | R-50 | R-101 | Rx-101 | D53 | D53 |
| mAP | 54.04% | 52.85% | 53.6% | 57.89% | **59.79%** |

FRC: Faster R-CNN, RN: RetinaNet, CRC: Cascade R-CNN, GRC: Grid R-CNN, FA: FreeAnchor, DRC: Dynamic R-CNN, SRC: Sparse R-CNN, VN: VarifocalNet, Y3: YOLOv3, Y3-T: YOLOv3+IWS (ours). R-50: Resnet-50, R-101: Resnet-101, D53: Darknet-53 Rx-101: Resnext-101.

### 5.1.2. Portability

Among YOLO series models, YOLOv5l (version 2021) is currently the best performing network model. Although YOLOv5l has not been published yet, it is still being updated and open sourced. After YOLOv5l was equipped with our IWS module, it can indicate that the IWS module is Portability and that the IWS module can indeed play a role if the indicator has risen. We used YOLOv5l to learn under IGBT_DF dataset as a new baseline. We found a problem here. The structure of YOLOv5l is different from that of YOLOv3 in the Neck and Head. When configuring the IWS module, YOLOv5l needed to be optimized and reconstructed, as is YOLOv5l-op. YOLOv5l-op can relatively and effectively maintain various indicators of YOLOv5l. As a result, there is a certain degree of improvement. After installing the IWS module on this basis, we found that all indicators improved. mAP has been increased by 3.1%. In mAP [0.5, 0.95], which is the more comprehensive indicator, there is still an increase of 2.87%. Under the IGBT_DF dataset, the IWS module is effective and performed relatively well (shown in Table 7). Similarly, we have optimized and added the IWS module in YOLOv3spp. It can be observed that under the premise of ensuring the detection speed, YOLOv3spp+IWS still has excellent performance. Acc of the IGBT board surface object detection task is up to the standard. In addition, with a 3.35% improvement in key indicator and a 6.54% improvement in similarities and differences identification indicator. In particular, mR has been significantly improved by 12.88%, reducing the missed detection rate (shown in Table 7). Experiments show that the IWS module can only be effective when it is mounted on a Head with multiple branches. Therefore, we recommend deploying the IWS module on the detection network with multi-head. Under the IGBT_DF_L dataset, the IWS module still performed well (shown in Table 8).

**Table 7.** Experimental results (more experimental results can be found in Table A1) of the IGBT_DF dataset based on YOLO series.

| Model | Acc | mP | mR | mAP | mAP [0.5, 0.95] | WAcc | FPS |
|---|---|---|---|---|---|---|---|
| YOLOv3 (baseline) | 89.86% | 45.16% | 67.18% | 50.98% | - | 76.13% | 35 |
| YOLOv3+IWS (ours) | 91.48% (+1.62%) | 52.58% (+7.42%) | 70.73% (+2.55%) | 53.12% (+2.14%) | - | 82.52% (+6.39%) | 34 |
| YOLOv3-spp (baseline) | 90.07% | 46.24% | 56.60% | 50.11% | - | 75.86% | 20 |
| YOLOv3-spp+IWS (ours) | 91.55% (+1.48%) | 46.08% (−0.16%) | 69.48% (+12.88%) | 53.46% (+3.35%) | - | 82.40% (+6.54%) | 31 |
| YOLOv5l (baseline) | 90.85% | 76.13% | 63.73% | 53.69% | 32.36% | - | 140 |
| YOLOv5l-op (ours) | 90.33% (−0.52%) | 79.89% (+3.76%) | 62.3% (−1.43%) | 54.67% (+0.98%) | 32.43% (+0.07%) | - | 132 |
| YOLOv5l+IWS (ours) | 91.71% (+1.37%) | 77.7% (+1.57%) | 65.1% (+1.37%) | 56.79% (+3.1%) | 35.23% (+2.87%) | - | 126 |

The value (the color mark) can be observed to increase and decrease compared with the corresponding baseline model.

**Table 8.** Experimental results of the IGBT_DF_L dataset based on YOLO series.

| Model | Acc | mP | mR | mAP | mAP [0.5, 0.95] | WAcc | FPS |
|---|---|---|---|---|---|---|---|
| YOLOv3 (baseline) | 91.71% | 43.12% | 74.8% | 57.89% | - | 84.72% | 35 |
| YOLOv3+IWS (ours) | 92.89% (+1.28%) | 46.66% (+3.54%) | 76.06% (+1.26%) | 59.79% (+1.9%) | - | 86.34% (+1.62%) | 34 |
| YOLOv3-spp (baseline) | 92.08% | 44.53% | 75.49% | 59.28% | - | 84.98% | 20 |
| YOLOv3-spp+IWS (ours) | 93.10% (+1.02%) | 45.02% (+0.49%) | 75.85% (+0.36%) | 59.45% (+0.17%) | - | 87.01% (+2.03%) | 31 |
| YOLOv5l (baseline) | 93.63% | 68.24% | 67.18% | 64.09% | 42.66% | - | 140 |
| YOLOv5l-op (ours) | 93.66% (+0.03%) | 67.17% (−1.07%) | 69.04% (+1.86%) | 64.08% (−0.1%) | 43.61% (+0.95%) | - | 132 |
| YOLOv5l+IWS (ours) | 93.89% (+0.26%) | 69.07% (+0.83%) | 69.76% (+2.58%) | 65.49% (+1.4%) | 44.46% (+1.8%) | - | 126 |

The value (the color mark) can be observed to increase and decrease compared with the corresponding baseline model.

### 5.1.3. Experiment Details for Each Class of Fractal Objects

We calculated the error detection rate of each network model for each class under the IGBT_DF_L dataset, which is the largest dataset. The result can be visualized in the form of heatmaps. We dealt with this to facilitate the observation of the network model's learn-ability for each class and to observe which classes the network model is prone to confusion. Figures 11 and 12 show that the error detection rate of network models equipped with the IWS module has a certain degree of decline.

$$e^d_{c(i),c(j)} = W_{c(i),c(j)} / C_{c(i),c(j)} \tag{13}$$

$$e^s_{c(i)} = W_{c(i)} / C^2_{c(i)}. \tag{14}$$

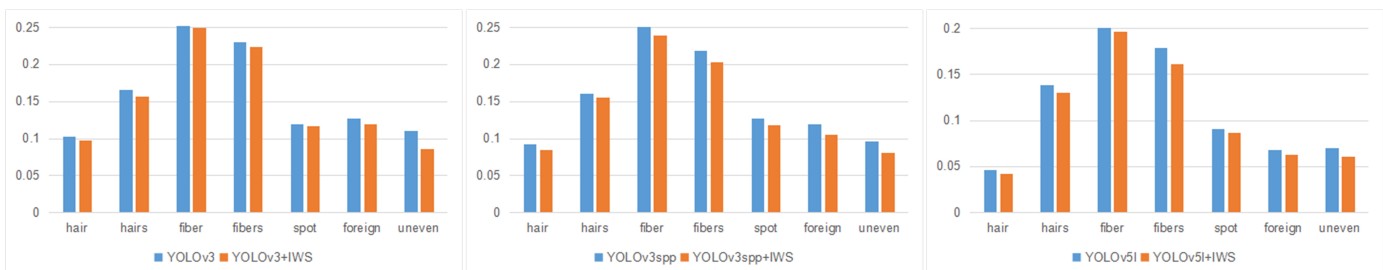

**Figure 11.** Visualization of the false detection rate of the same class as different classes.

Figure 11 reflects the proportion of objects of the same class that are incorrectly detected as different classes. This part is similar to recall, but it is more suitable for error detection rate statistics than recall is. In Figure 11, we only care about the error detection rate ($e^s_{c(i)}$ in Equation (14)) of each class, and do not focus on the specific class of error detection. The statistical function is defined in Equation (14). Figure 12 reflects the proportion of different classes of objects being wrongly detected as the same class. Figure 12 describes the scene where the real class of detection object A and detection object B are different (assuming the true class label of A is $c(i)$ and the true class label of B is $c(j)$), but A is incorrectly detected and regarded as $c(j)$. In Figure 12, we focus on the error detection rate ($e^d_{c(i),c(j)}$ in Equation (13)) between classes, which reflects the degree of confusion between various classes of the detection network model. The statistical function is outlined as Equation (13). We used the contour surface map from the top view to show the contrast effect. Combining the color change of the top view surface and the size of the area contained in the contour line, the size change of the false detection rate is determined. The smaller the contained area of the contour line under the same color, the smaller the error rate. The lower the color index on the right side of Figure 12, the lower the error rate.

As an example, the coordinates (fibers, uneven) in Figure 12 represent the probability that YOLOv5l+IWS is used as a detection network model to detect "fibers" as "uneven". Observing the part of the yellow circular dashed frame in Figure 12, the error detection rate of the detection network model optimized with the IWS module is less than the error detection rate of the baseline model. Our detection network model considerably reduces the false detection rate of two pairs of easily confusing classes (refer to Figure 13). One pair is "fibers" and "uneven". The other pair is "spot" and "foreign".

It is not difficult to see that "fiber" and "fibers" are the most easily misdetected group of classes. The two are a pair of challenging confusion classes. This pair of fractal object detection groups is much more difficult than the previous two groups. If observed in the coordinate system of the same range, it is difficult to see the change. Consequently, we put these more difficult fractal object detection groups separately for observation. As can be seen from Figure 14, for the more difficult fractal object detection groups, the IWS module can still play a role. Error detection rates are reduced to varying degrees. The experimental results show that the IWS module does reduce the detection error rate of fractal objects by adding one more analysis dimension to object information by enhancement learning. Based on various laboratory data, YOLOv5l equipped with the IWS module performs relatively well. Therefore, we referred to this version of the foreign object detection network model as YOLO+IWS. Under the two scales of datasets, the performance of YOLO+IWS is not only reflected in the improvement of overall indicators, but also in Recall of each class (refer to Figure 15).

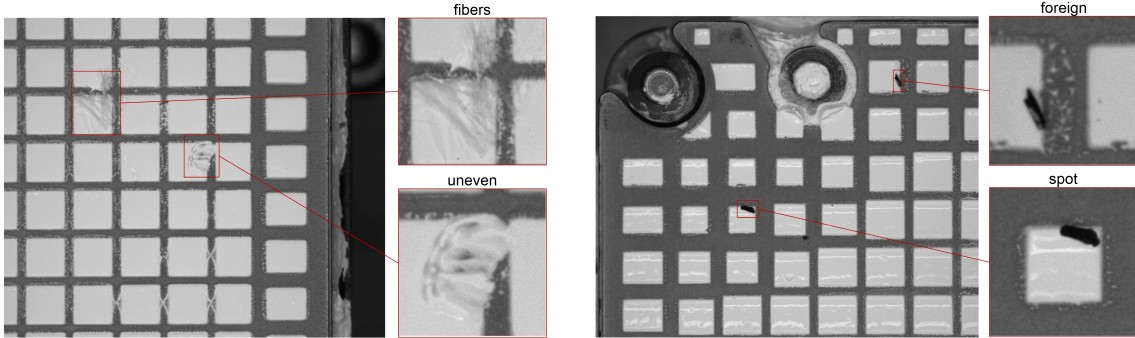

**Figure 12.** Visualization of the false detection rate of different classes as the same class.

**Figure 13.** Two pairs of easily confusing classes.

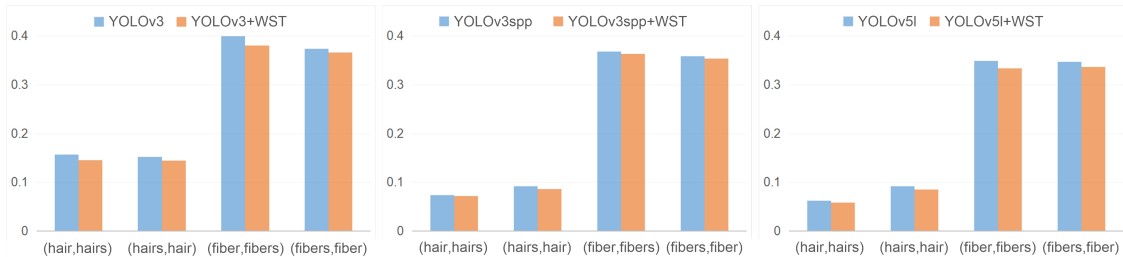

**Figure 14.** The hardest fractal object detection group (fiber and fibers) and the second-hardest fractal object detection group (hair and hairs) in Figure 12 are presented in the form of histograms.

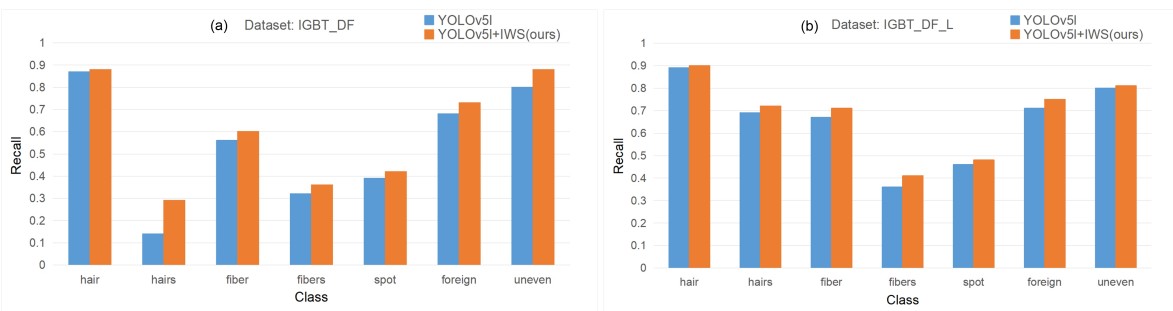

**Figure 15.** The comparative performance of each class under different scale datasets.

### 5.2. On-Site Detection

In the traditional method (computer vision detection method based on Open CV), if the threshold is set too small, it will cause the detection to be too sensitive and easy to false alarms. If the threshold is not small enough, it is easy to miss the detection. In addition, the traditional method only performs binary classification operations, and cannot obtain and collect object information. Choosing to use deep learning detection methods is to solve these problems. What we have done is to solve difficulties encountered by deep learning detection methods in the application of IGBT board surface object detection, which is our research focus. We have provided optimization schemes.

As shown in Figure 16, YOLO+IWS (see Figure 16c) is more accurate than YOLOv5l object detection algorithm (see Figure 16b) for the detection of fibers with relatively low contrast in the detection of IGBT board surface objects. YOLOv5l's missed detection problem (see Figure 16e) has been effectively improved in YOLO+IWS (see Figure 16f). For more on-site comparison of actual detection results, see Figure A1.

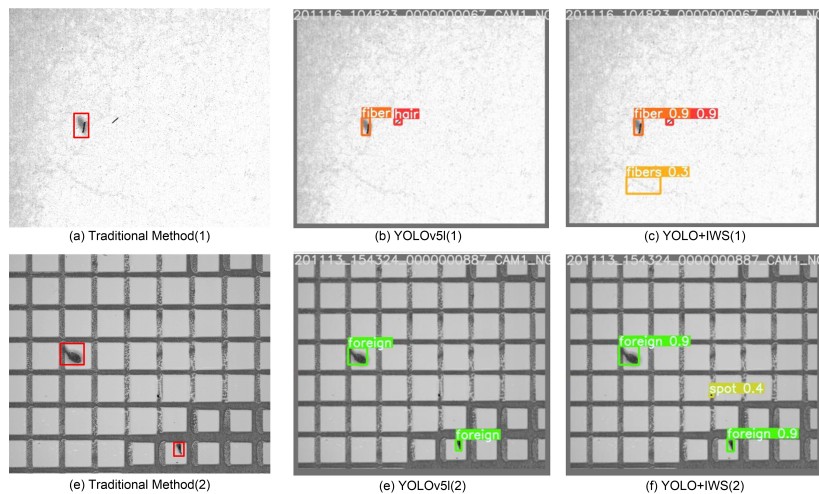

**Figure 16.** Visualization effect contrast.

## 6. Application in Object Detection of Real-World IGBT Coating Operation

We mount our YOLO+IWS object detection network model in the IGBT automatic glue detection and tightening production line server to complete the realization of the IGBT board surface object detection function (refer to Figure 17). This forms a complete IGBT board surface object detection system. This realizes the digital transformation of detection equipment and provides strong support for the subsequent adjustment and optimization of control decision-making. The network model is encapsulated, and the front-end interface is formed through the Flask and Vue frameworks to realize interaction.

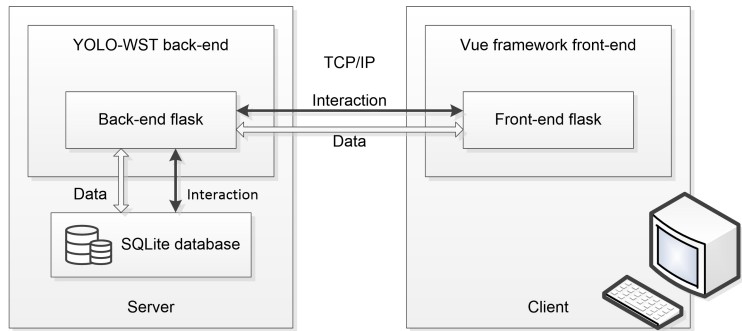

**Figure 17.** Software architecture.

The real IGBT automatic glue coating detection and tightening production line is shown in Figure 18, which mainly includes IGBT automatic glue coating equipment, IGBT board surface object detection equipment, and IGBT tightening links. On-site staff can observe the operation status in real-time through the display. The production line has an independent server to provide intelligent manufacturing requirements, such as PLC control, algorithm model triggering, and data integration processing. As shown in Figure 19, the real-time foreign object detection visualization results of images are in the production line (see Figure 19 yellow box No. 4). Qualitative and quantitative information about the detected foreign objects is displayed here (see Figure 19 yellow box No. 1). There is a display of the cumulative amount of detection results for the foreign object class (see Figure 19 yellow box No. 2), which provides a basis for the production control plan of the production line. This also provides a basis for the learning strategy of the foreign object class for the network model's incremental learning. We display indicators after incremental learning of the network model at each stage to reflect the performance status of the network model (see Figure 19 yellow box No. 3).

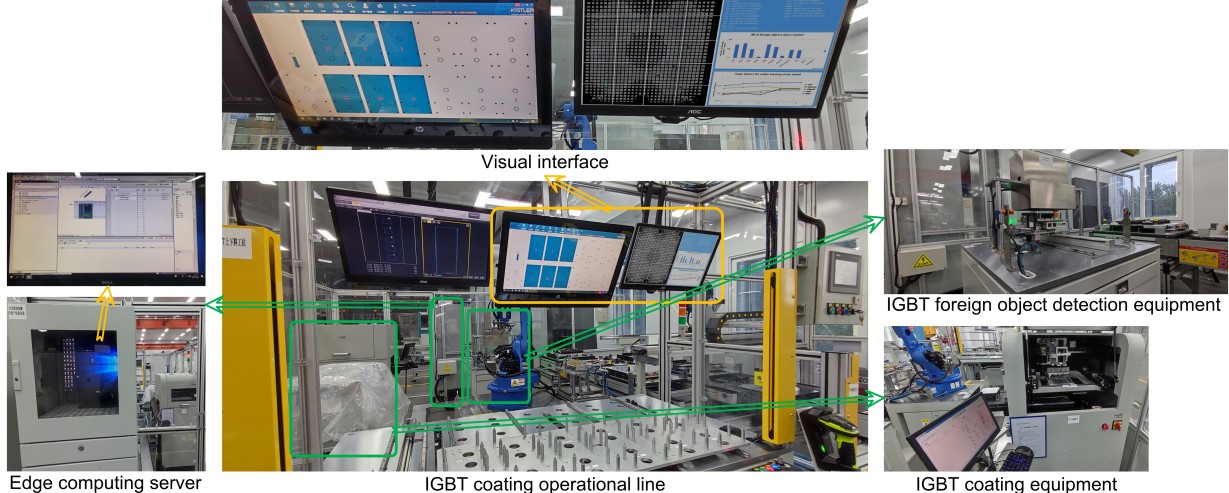

**Figure 18.** IGBT automatic glue coating detection production line.

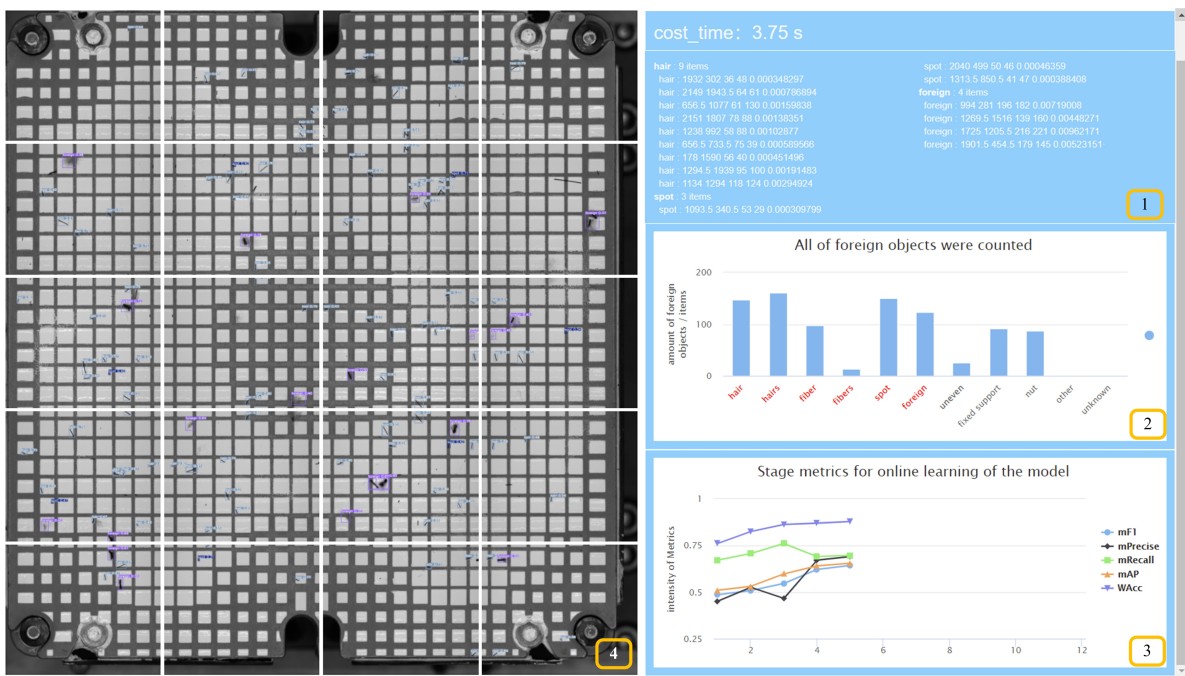

**Figure 19.** IGBT board surface object detection visual interface.

## 7. Conclusions

We chose detection models of the YOLO series as the base model framework, which is a more common detection model in industrial visual inspection. To improve the fractal object detection capability in industrial vision detection, we proposed YOLO+IWS, an end-to-end, easy-to-deploy, and easy-to-learn object detection model. It can collect more detailed and comprehensive detection information, which is helpful for enterprises to make better decisions.

Experimental results show that both the IWS module and optimization strategies perform well. YOLO+IWS has improved the key indicator. Not only that, we applied the IWS module and optimization strategies to form a holistic approach to object detection. It allows decision-makers to observe the results of detection in real time and to revisit historical data at any time as needed.

In the future, our research will focus on fractal object detection on few-shot learning in industrial vision inspection tasks. It is still considered to ensure real-time and accurate detection on the basis of existing detection capabilities and low resource costs.

**Author Contributions:** Conceptualization, H.H. and X.L.; methodology, H.H.; software, H.H.; validation, H.H.; formal analysis, H.H.; investigation, H.H. and X.L.; resources, X.L.; data curation, H.H.; writing—original draft preparation, H.H.; writing—review and editing, H.H. and X.L.; visualization, H.H.; supervision, H.H.; project administration, H.H. and X.L.; funding acquisition, X.L. All authors have read and agreed to the published version of the manuscript.

**Funding:** This research was funded by National Key R&D Program of China (2019YFB1705002, 2017YFB0304100), National Natural Science Foundation of China (51634002), LiaoNing Revitalization Talents Program (XLYC2002041), and the Open Research Fund from the State Key Laboratory of Rolling and Automation, Northeastern University, (Grant No.: 2018RALKFKT008).

**Institutional Review Board Statement:** Not applicable.

**Informed Consent Statement:** Not applicable.

**Data Availability Statement:** The source codes and datasets used to support the findings of this study are available from the corresponding author upon request via email: royhh1990@163.com.

**Conflicts of Interest:** The authors declare no conflict of interest.

## Abbreviations

The following abbreviations are used in this manuscript:

FGI      Fine-Grained Information
WST    Watch and Study Tactic
IWS     Information Watch and Study
MRD    Mutli-Task Result Discriminant
WAcc   WST Accuracy

## Appendix A

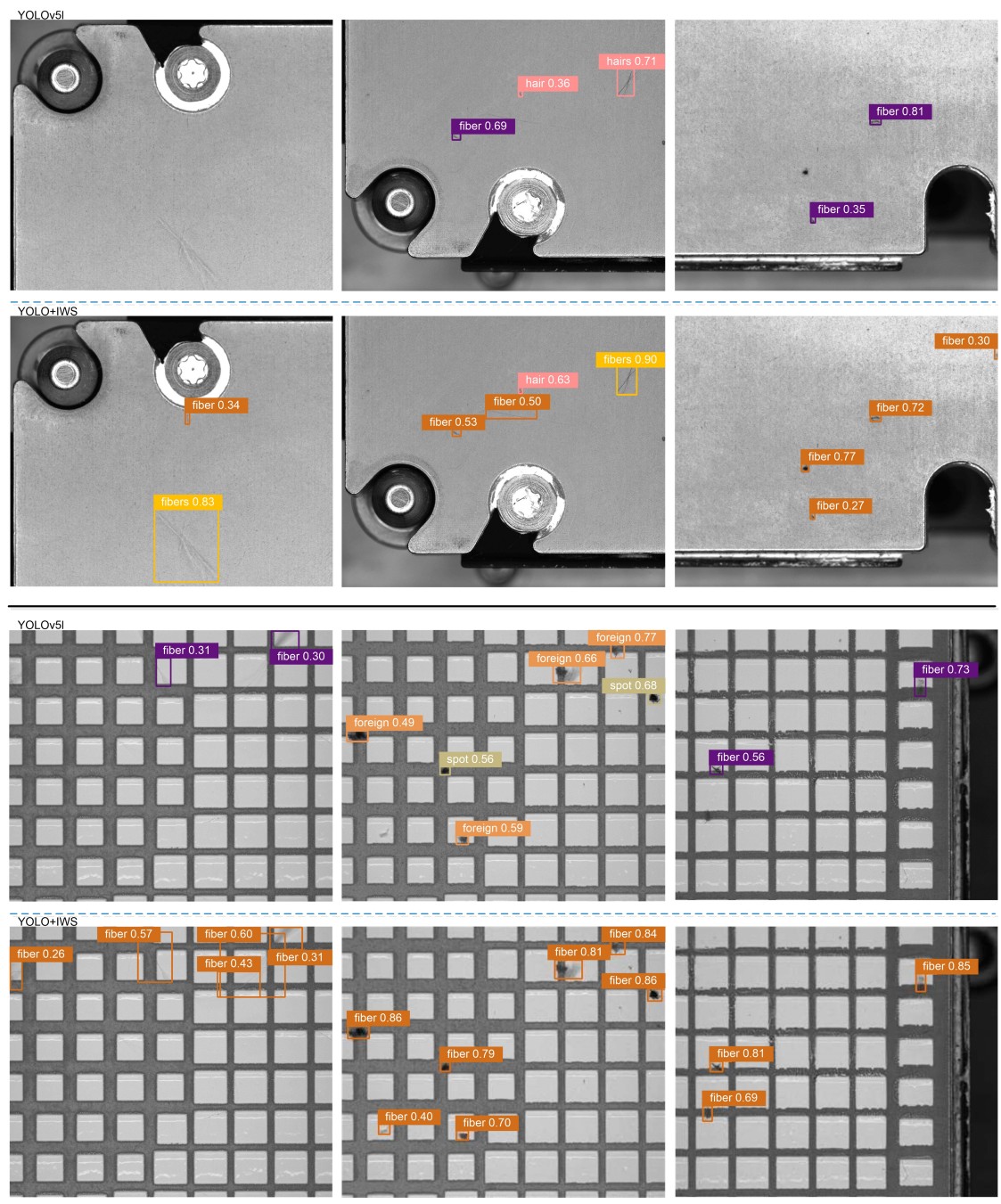

**Figure A1.** Visualization of detection results using YOLOv5l and YOLO+IWS in the field.

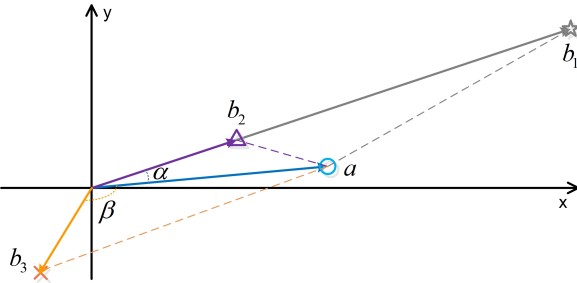

**Figure A2.** Taking a two-dimensional space as an example, count the metrics from $b_1$, $b_2$, and $b_2$ to $a$, respectively. The Cosine metric cannot reflect the difference between $b_1$ and $b_2$ for $a$. The L2 metric cannot reflect the difference between $b_1$ and $b_3$ for $a$.

## Appendix B

*Appendix B.1. More Experimental Results*

**Table A1.** Experimental results of IGBT_DF dataset based on YOLOv7 and YOLOv7x.

| Model | Acc | mP | mR | mAP | mAP [0.5, 0.95] |
|---|---|---|---|---|---|
| YOLOv7 (baseline) | 91.56% | 73.74% | 68.99% | 54.02% | 33.88% |
| YOLOv7+IWS (ours) | 92.84% | 79.35% | 68% | 57.02% | 35.97% |
| YOLOv7x (baseline) | 91.53% | 69.34% | 70.27% | 53.86% | 33.46% |
| YOLOv7x+IWS (ours) | 92.55% | 69.79% | 69.76% | 58.89% | 37.42% |

*Appendix B.2. Ablation Study*

The Y point is the input node of the YOLO Head. The position of point P is shown in Figure 6. Referring to Table A2, our theory in Section 3.1.1 is verified. FGI is a very suitable extraction point. The experimental results under the comparison of different measurement methods are as follows. Our theory in Section 3.1.2 is verified. CE is suitable and effective. Referring to Table A2.

**Table A2.** Comparison of different extraction points and different the measure.

| Model | EP | WM | mP | mR | mAP |
|---|---|---|---|---|---|
| YOLOv3 | Y | - | 43.12% | 74.80% | 57.89% |
| Ours | Y | CE | 44.88% | 75.61% | 58.55% |
| | P | CE | 45.25% | 75.11% | 58.66% |
| | FGI | Sim | 46.12% | 75.12% | 59.03% |
| | FGI | CE | 46.66% | 76.06% | 59.79% |

EP: extraction points for fine-grained information. WM: the measure of the WST learner.

*Appendix B.3. The Specific Information of IGBT_DF_L Dataset*

**Table A3.** The specific information of IGBT_DF_L dataset.

| IGBT_DF_L | Image | Object | Specific Information (Objects) | | | | | | |
|---|---|---|---|---|---|---|---|---|---|
| | | | Hair | Hairs | Fiber | Fibers | Spot | Foreign | Uneven |
| Train subset | 3544 | 18,139 | 4587 | 233 | 2321 | 467 | 4562 | 5022 | 947 |
| Test subset | 886 | 4512 | 1146 | 145 | 478 | 126 | 1110 | 1274 | 233 |

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
