# Peer review of "A Holistic Approach to IGBT Board Surface Fractal Object Detection Based on the Multi-Head Model"

_machines, doi:10.3390/machines10080713_

Round 1

Reviewer 1 Report

This paper proposes a holistic approach to fractal object detection based on a multi-head model with object information enhancement learning. This paper has different contributions concerning the IWS module to increase the detection dimension, the FGI module to extract more comprehensive feature vectors, and the MRD strategy to give detection module. 

The proposed algorithm's efficiency is tested on a dataset from Beijing Zongheng Electromechanical Co., Ltd.   Experimental results show the efficiency of the proposed model.

The proposed architecture is interesting. Furthermore, the test results for this architecture are also encouraging. Despite promising results, the framework presented in this paper still has issues. Below are some comments intended to aid the authors in improving this paper.

1) The research has compared the proposed methods with YOLOv5 or YOLOv3 as a baseline that is not the latest version of YOLO. Why did the authors not compare with more recent versions such as YOLOv7?

2) Why did the authors only test the proposed method with a private dataset? It is also better to investigate with a publicly recognized dataset for one module or whole system to check the efficiency of the authors' approach. It is also difficult for the third party to redo the experiments.

3)The reviewers also want to know how the authors divide the dataset into train/test subsets. How can the authors validate that the classes are correctly labeled in the dataset?

4) The parameter settings of authors' YOLOv3-based and YOLOv5-based versions in experiments are also important. How can we adapt from YOLOv3 to YOLOv5?

Author Response

August 9, 2022

Dear Reviewer,

Thank you for your comments concerning our manuscript entitled “A Holistic Approach to IGBT Board Surface Fractal Object Detection Based on the Multi-Head Model” (ID: machines-1813933). Those comments are all valuable and very helpful for revising and improving our paper, as well as the important guiding significance to our researches.

We have studied comments carefully and have made our point-by-point responses which we hope meet with approval. Revised portion are marked in orange in the resubmitted manuscript.

Please see the attachment (Response reviewer 1- MDPI.pdf).

Reviewer 2 Report

The article proposes a Holistic Approach to IGBT Board Surface Fractal Object Detection Based on the Multi-Head Model. However, there are many problems in the article.

Please confirm the grammar and spelling in detail. There is some incorrect use of punctuation.

Please change the reference order of figures, such as “Figure11 and Figure12” in 5.1.3.

The traditional method in Figure 16 is a typo.

The author's name in the citation of 46 is incorrect.

Details of the design should be described in the introduction.

The structure of the paper is terrible.

The paragraph setting in this article is unreasonable, and the logic between paragraphs is not smooth.

For example, lots of the paragraphs only have one or two sentences.

The related work (section 2) should be divided into several sections ().

The author proposes that the two-stage object detection model is not suitable for industrial applications, please explain in detail why it is not suitable?

Later, the paper introduces that yolov3 is widely used in industrial visual inspection, and the front and back logic should be unified.

Figures 6, 9, Tables 2, 3, etc. are not cited in the article.

Please describe the dataset components in detail in the dataset introduction section, e.g. refer to Table 2.

Please show the input size of the models. Please shows the parameter setting of the confidence and nms-IoU.

Why are the backbone structures used in comparison models in Table 4 ResNet50 and ResNet101?

What are the results of the original backbone structure?

Please, introduce the traditional methods which are shown in subsection 5.2?

Also, I would like to know how the traditional method is to determine the target location information and frame it. Please describe the action in the response letter. 

Author Response

August 9, 2022

Dear Reviewer,

We would like to thank you for the opportunity to revise and resubmit our manuscript entitled “A Holistic Approach to IGBT Board Surface Fractal Object Detection Based on the Multi-Head Model” (ID: machines-1813933). We found your comments to be helpful in revising the manuscript and have carefully considered and responded to each suggestion, corresponding changes to the resubmitted manuscript are marked in orange.

We have responded to all the comments mentioned by reviewers. We hope that these modifications will get your approval.

Please see the attachment (Response Letter 2- MDPI.pdf).

Round 2

Reviewer 1 Report

I think this paper now is suitable for publication

Reviewer 2 Report

The annotations in Figure 4 are ambiguous.